# Mucoadhesive Poloxamer-Based Hydrogels for the Release of HP-β-CD-Complexed Dexamethasone in the Treatment of Buccal Diseases

**DOI:** 10.3390/pharmaceutics13010117

**Published:** 2021-01-18

**Authors:** Raul Diaz-Salmeron, Balthazar Toussaint, Nicolas Huang, Etienne Bourgeois Ducournau, Gabriel Alviset, Sophie Goulay Dufaÿ, Hervé Hillaireau, Amélie Dufaÿ Wojcicki, Vincent Boudy

**Affiliations:** 1Département de Recherche et Développement Pharmaceutique, Agence Générale des Equipements et Produits de Santé (AGEPS), 75005 Paris, France; balthazar.toussaint@aphp.fr (B.T.); ebourgeois@septodont.com (E.B.D.); gabriel.alviset@unither-pharma.com (G.A.); sophie.dufay@aphp.fr (S.G.D.); amelie.wojcicki@aphp.fr (A.D.W.); vincent.boudy@aphp.fr (V.B.); 2CNRS UMR 8258—Inserm U1022, Paris Descartes University, 75006 Paris, France; 3Institut Galien Paris Saclay, Université Paris-Saclay, CNRS, 92296 Châtenay-Malabry, France; nicolas.huang@universite-paris-saclay.fr (N.H.); herve.hillaireau@universite-paris-saclay.fr (H.H.)

**Keywords:** mucoadhesion, controlled-drug release, polysaccharides, poloxamers, cyclodextrins

## Abstract

Oral lichen planus (OLP) is an ongoing and chronic inflammatory disease affecting the mucous membrane of the oral cavity. Currently, the treatment of choice consists in the direct application into the buccal cavity of semisolid formulations containing a corticosteroid molecule to decrease inflammatory signs and symptoms. However, this administration route has shown various disadvantages limiting its clinical use and efficacy. Indeed, the frequency of application and the incorrect use of the preparation may lead to a poor efficacy and limit the treatment compliance. Furthermore, the saliva clearance and the mechanical stress present in the buccal cavity also involve a decrease in the mucosal exposure to the drug. In this context, the design of a new pharmaceutical formulation, containing a steroidal anti-inflammatory, mucoadhesive, sprayable and exhibiting a sustained and controlled release seems to be suitable to overcome the main limitations of the existing pharmaceutical dosage forms. The present work reports the formulation, optimization and evaluation of the mucoadhesive and release properties of a poloxamer 407 thermosensitive hydrogel containing a poorly water-soluble corticosteroid, dexamethasone acetate (DMA), threaded into hydroxypropyl-beta-cyclodextrin (HP-β-CD) molecules. Firstly, physicochemical properties were assessed to ensure suitable complexation of DMA into HP-β-CD cavities. Then, rheological properties, in the presence and absence of various mucoadhesive agents, were determined and optimized. The hydration ratio (0.218–0.191), the poloxamer 407 (15–17 wt%) percentage and liquid-cyclodextrin state were optimized as a function of the gelation transition temperature, viscoelastic behavior and dynamic flow viscosity. Deformation and resistance properties were evaluated in the presence of various mucoadhesive compounds, being the sodium alginate and xanthan gum the most suitable to improve adhesion and mucoadhesion properties. Xanthan gum was shown as the best agent prolonging the hydrogel retention time up to 45 min. Furthermore, xanthan gum has been found as a relevant polymer matrix controlling drug release by diffusion and swelling processes in order to achieve therapeutic concentration for prolonged periods of time.

## 1. Introduction

Lichen planus (LP) is a chronic inflammatory dermato-mucosal disease affecting most frequently the skin and oral mucosa areas, and particularly involving the buccal mucous membrane. The pathology is called buccal or oral lichen planus (BLP or OLP). The nature and origin of OLP are still unknown but various theories concerning its etiology have been proposed in the past [1,2]. That includes bacteria [3] or viral origin [4], neurogenic theories [5], environment or life-style factors [6] and other hypothesis associated with other diseases.

Currently, the treatment of choice for the OLP consists of the use of topical steroids. The use of many topical steroid compounds is widely described in the scientific literature. Thus, betamethasone [7], fluocinonide [8], hydrocortisone hemisuccinate [9], fluticasone [10] and other therapeutic molecules such as mometasone furoate [11], clobetasol propionate [12,13,14,15] or triamcinolone [16] demonstrated an improvement of signs (erythema, reticulation and ulceration) [17] and symptoms (pain and discomfort) [18].

However, existing topical steroid treatment has shown various disadvantages and limitations restricting their clinical use. Indeed, the frequency of application of corticosteroids limits the treatment compliance and makes it difficult to reach optimal effects. Thus, to achieve correct concentrations, the treatment may be applied between five to ten times daily [19], leading often to the ingestion of pharmaceutical preparations following incorrect use of the preparation [20]. Furthermore, the elderly, considered as a significant part of patients suffering OLP, found technically difficult to apply drug treatment to all locations of the buccal cavity, which means a lack of full contact between the drug and affected areas [21]. In addition, young people, with their current active lifestyle, need an effective and easy-to-apply treatment.

The permanent washing out of the buccal cavity by saliva also limits the average length of mucosal exposure to a drug treatment by three main mechanisms: drug dilution into the oral cavity, saliva clearance and mechanical stress due to swallowing and mastication processes [22].

In recent years, many dosage forms have been designed for buccal drug delivery in order to overcome the main limitations of this administration route [23,24,25,26,27,28,29]. Some interesting research work has been developed on thermosensitive poloxamer-based hydrogels, particularly with poloxamer 188 (P188) and 407 (P407). Indeed, thermosensitive hydrogels for buccal drug delivery have been designed and described in scientific literature [29,30,31,32,33]. These kind of hydrogels can undergo a sol–gel transition depending on the temperature [34,35]. Their chemical structure enables the self-association of hydrophobic poly(propylene oxide) (PPO) regions as micelles in an endothermic process [36,37], and subsequently the gelation, due to the formed micelle interactions, to form an elastic and three-dimensional network (solid-like). Thus, poloxamer-based hydrogels, and particularly P407, due to its colorless and washable properties [36], could be liquid at room temperature (20–25 °C) and is usually gelling between 30 and 35 °C. These thermosensitive properties make P407-based hydrogels appropriate for their use in buccal spraying devices [38] in order to reach an in situ gelation [39]. Additionally, these hydrogels are the most commonly used thermosensitive system in the pharmaceutical field [40,41,42,43] because they are not irritant for the skin and the mucous membranes. Nevertheless, poloxamer-based hydrogels exhibited very poor mucoadhesive properties once placed in the biological medium. Many strategies such as chemical modifications in poloxamer gels can be attempted. Thus, chemically-modified poloxamer gels have been already described in the literature to obtain improved mucoadhesive and mechanical properties [44,45]. Furthermore, physical strategies can be intended to reach acceptable mucoadhesive properties. Therefore, mucoadhesive agents or polymers have been used to improve the adhesive behavior of hydrogels. Anionic polymers, such as polyacrylic acid (PAA) and its derivatives (trade name carbomer) have been widely used because they exhibit excellent mucoadhesive characteristics due to the formation of strong hydrogen bonding interactions with the oligosaccharide chains of mucin [46,47,48]. Alginate, a naturally linear polysaccharide extracted from seaweed, bacteria and algae, has been also used for the preparation of mucoadhesive hydrogels [49,50]. In particular, high-molecular weight alginate, due to its flexibility properties, is able to bridge distant mucin sites causing the protein contraction and its mucoadhesive effect [48,51]. Xanthan gum (XG), another anionic polysaccharide, is widely used in the pharmaceutical and cosmetic industry for its mucoadhesive properties [25,52]. This polysaccharide also generates a viscous medium slowing drug release in sustained release formulations [53,54,55].

Since the management of OLP treatments can vary considerably due to the lack of official guidelines [20], the treatment may be adjusted in the function of the disease stage for individual patients [56,57]. Dexamethasone, a class II steroidal anti-inflammatory, less potent than clobetasol but with less side effects, could be used for this purpose. Indeed, a drug delivery improvement of a dexamethasone formulation may be particularly useful for its clinical use even for other pathologies than OLP.

In this context, the aim of this experimental study was to formulate a hydrophobic and practically insoluble in water API [58,59], dexamethasone acetate (DMA), widely known for its anti-inflammatory properties [60,61,62], into a thermosensitive P407-based hydrogel containing different mucoadhesive agents, such as carbomer, sodium alginate or xanthan gum, to improve their interactions with mucins present at the buccal surface mucosa.

The main novelty of this study was the use of cyclic oligosaccharides to improve DMA solubility into a hydrogel. Due to the hydrophobic nature of DMA, the solubilization into an aqueous hydrogel is not adapted. To solve this problem, DMA molecules were previously complexed into the cavity of cyclodextrin molecules (CD), particularly into HP-β-CD, the most successful β-CD derivative to date [59,63,64]. CD compounds, produced by enzymatic degradation of starch [65] are able to form inclusion complexes with water-insoluble API due to different supramolecular forces such as Van der Waals or hydrophobic interactions [66]. CD molecules were of interest given the presence of hydrophilic groups on the outside, which is useful for maintaining the hydrophilicity and swelling behavior of the hydrogel. Their hydrophobic cavity enables and improves the entrapment, the apparent solubility and the controlled-release of hydrophobic drugs [34].

This paper describes for the first time the design of a novel thermosensitive hydrogel for buccal drug delivery intended to be sprayable (presenting a dynamic viscosity of less than 200 mPa.s at room temperature in order to suit for the identified device), and exhibiting an optimized sol–gel transition temperature (30–32 °C) that could allow the in situ gelation once administered in the oral cavity for the treatment of inflammatory diseases. Furthermore, satisfactory mucoadhesive properties and sustained and controlled drug release were obtained. In this paper, we focused on the formulation process of hydrogels and the mechanism of DMA encapsulation into HP-β-CD molecules and the rheological characterization of hydrogels and sustained-release. This paper describes for the first time the use of a CD complexed anti-inflammatory active substance contained in a mucoadhesive polymer matrix for the treatment of buccal diseases as OLP. Designing a specific formulation for the oral mucosa should improve the treatment efficiency due to a better absorption and the patient compliance due to an appropriate formulation.

## 2. Materials and Methods

### 2.1. Materials

Dexamethasone acetate was purchased from Aventis (Gentilly, France). Kolliphor-P407 was from BASF (Mississauga, Canada). Hydroxy-propyl-beta-cyclodextrin was purchased from Roquette (Paris, France). Mucoadhesive agents used in the experiments were xanthan gum (Satiaxane from Cargill, Puteaux, France), carbomer (Carbopol C971P from Lubrizol, Brussels, Belgium) and sodium alginate (Protanal LF 10/60 FT from FMC Biopolymer, Cork, Ireland). Porcine gastric mucin type II was from Sigma Aldrich (Saint-Louis, MO, USA). All the experiments were performed using sterile water from Fresenius Kabi (Sévres, France).

For the HBSS medium preparation, the chemical products used were calcium chloride × 2H_2_O, potassium chloride, potassium hydrogen phosphate anhydrate, sodium chloride, disodium dihydrogen phosphate × H_2_O, sodium hydrogen carbonate, magnesium chloride × 6H_2_O and magnesium sulfate × 7H_2_O. All these products were purchased from Sigma Aldrich (Saint-Louis, MO, USA).

### 2.2. Methods

#### 2.2.1. Preparation of Hydrogels

Hydrogels were prepared according to the method called the “cold method” [36,67]. Concentrations of all components reported in this article were expressed as the weight/weight ratio (%*w*/*w*). The formulations were prepared following two different approaches. In both approaches the strategy was to form inclusion complexes between DMA and HP-β-CD in order to increase the apparent solubility of DMA. Indeed, its low aqueous solubility is limiting its clinical usefulness [68].

The first approach consisted of solubilization under magnetic stirring at 400 rpm at 4 °C of amounts of P407 (15–17% weight/weight (*w*/*w*)) in 20 g of distilled water containing the corresponding mucoadhesion agent previously dissolved (at 0.1 and 0.5% *w*/*w* for Carbopol C971P and sodium alginate and 0.05 and 0.1% *w*/*w* for the xanthan gum). At the same time, a second aqueous solution was prepared by adding amounts of HP-β-CD and DMA (0.1% *w*/*w*) in 30 g of distilled water under magnetic stirring at 400 rpm at room temperature. The first solution containing the dispersed amount of P407 was completed with the total volume of the second aqueous solution. The final solution was allowed to stir at 400 rpm for 24 h at 4 °C in order to stay in the solution state of the thermogelling agent P407 and not to disadvantage the homogeneity of the preparation. After 24 h, the total formulation weight was adjusted to 100 g by adding distilled water and stirring for 5 min.

For the second approach, mucoadhesion agent was totally dissolved in 20 g of sterile water under magnetic stirring at 400 rpm. Then, the corresponding amount of P407 was progressively and gradually added. The final weight of the aqueous solution was completed with 20 g of sterile water. This solution was allowed to rest at 4 °C for 24 h. A second aqueous solution was prepared by dissolving amount of HP-β-CD in 30 g of sterile water under magnetic stirring at 400 rpm during 5 min. For the formulations containing the API (0.1% *w*/*w*), the exact amount of DMA was added and allowed to stir for 1 h at 400 rpm then ultrasonicated during 2 min.

Once, the formulation approach influence was investigated, the first method was retained for the characterization of physicochemical properties of hydrogels and in vitro dissolution studies. Table 1 summarizes quantitative and qualitative composition of hydrogels tested.

#### 2.2.2. Phase Solubility Diagrams and Apparent Solubility Determination

Phase solubility studies were carried out according to the method described by Higuchi and Connors [69]. An excess of DMA (50 mg corresponding to 0.1% *m*/*v*) was added in a glass vial containing 50 mL of an aqueous solution of HP-β-CD at concentrations ranging from 0.46 to 23 mM representing 1/0; 1/0.2; 1/0.4; 1/1; 1/2; 1/4 and 1/10 molar ratios (DMA/HP-β-CD). Vials were kept under magnetic stirring at 400 rpm at room temperature for 72 h (duration estimated to be enough to reach the complexation equilibrium). The intrinsic solubility of DMA in sterile water was determined using the same protocol without the presence of cyclodextrin molecules. Then the samples were filtered through a 0.22 µm membrane filter and diluted 1/20 for UV spectrophotometric determination at a wavelength of 242 nm. Absorbance of different sample ratios was transformed in molar concentrations using a standard curve.

The standard curve was prepared in an aqueous solution of DMA/HP-β-CD using a stock solution containing 1.15 mM (25 mg) and 19.2 mM (1350 mg) of DMA and HP-β-CD respectively. The concentration of DMA should be below the solubility of the inclusion complexes. The stock solution was diluted 1/20 with sterile water then diluted again to obtain valid concentrations for quantification ranging from 0.058 to 0.012 mM. In order to evaluate the matrix effect of HP-β-CD, another standard curve using a stock solution of DMA in propylene glycol (PG) was prepared in the same range of concentrations because of the total solubility of DMA in PG. The equations of the calibration curves were y = 13.492x + 0.0126, r^2^ = 0.999 and y = 15.046x − 0.0065, r^2^ = 0.998 in an aqueous solution of HP-β-CD and PG, respectively. 

Slope values of each calibration curve were compared to investigate this effect.

#### 2.2.3. Rheological Characterization of Formulations

All the rheological studies were carried out on an Anton Paar MCR102 Rheometer (Graz, Austria). The geometry chosen was a stainless steel cone/plate equipped with a solvent trap (diameter 50 mm, angle 1° and truncation 100 µm) providing a homogeneous shear of the samples. All the data obtained were analyzed using the Anton Paar RheoCompass™ software version 1.25 (Graz, Austria) associated to the rheometer. All experiments were performed in triplicate and all data are expressed as the mean ± standard deviation.

##### Flow Rheometry Studies

Continuous shear analysis of formulations was performed at 25 and 37 °C using a stress sweep. Samples were carefully placed onto the inferior plate and allowed to equilibrate for at least 2 min prior to analysis. Flow curves were measured over shear rates ranging from 0.1 to 1000 s^−1^. The shear rate was firstly increased over a period of 60 s (upward curve), held at the upper limit for 30 s and subsequently decreased gradually to the lower limit over a period of 60 s (downward curve).

In each case, the continuous shear properties of at least three replicates were determined and the descending flow curves were fitted to a Carreau–Yasuda model equation [70] as follows:(1)$$ƞ=\;{ƞ}_{\infty }+\;\left({ƞ}_{0}-{ƞ}_{\infty }\right)\times {\left[1+\;{\left(\lambda \dot{\gamma }\right)}^{a}\right]}^{\left(n-1\right)/a}$$
where ƞ is the viscosity, $\dot{\gamma }$ is the shear rate, ƞ_0_ is the limiting steady state viscosity at the zero shear rate, ƞ_∞_ is the limiting steady state viscosity at the infinite shear rate, *n* is the power index and confers a power-law behavior to the curve, *a* is the Yasuda exponent describing the transition from the first Newtonian plateau to the power-law behavior and *λ* is the time constant setting the position of the power-law behavior as a function of the shear rate.

##### Oscillatory Rheometry Studies and Determination of Gelation Transition Temperature

A sinusoidal shear was applied to the sample in order to determine the storage (or elastic) modulus *G′* and the loss (or viscous) modulus *G″*. The stress *τ* (*t*) and the strain *γ* (*t*) were defined as follows in the Equations (2) and (3): (2)$$\tau \;\left(t\right)=\;\tau_{0}\cos\left(\omega t\right)$$
(3)$$\gamma \;\left(t\right)=\;\gamma_{0}\cos\left(\omega t-\;\delta \right)$$
where *τ_0_* and *γ_0_* are the maximal amplitudes of stress and strain respectively, *ω* is the shear pulsation (*ω = 2πN*; *N* = frequency) and *δ* is the phase angle stress/strain. 

The phase angle between stress and strain will define the storage (*G′*) and the loss (*G″*) moduli as follows in Equations (4) and (5):(4)$$G^{\prime }=\;\frac{\tau_{0}}{\gamma_{0}}\cos\delta$$
(5)$$G^{\prime \prime }=\;\frac{\tau_{0}}{\gamma_{0}}\sin\delta$$

After performance of an amplitude sweep (*f* = 1 Hz) and a frequency sweep (*γ* = 0.1%), the linear viscoelastic region (LVR) was estimated. All subsequent measurements of the storage modulus (*G′*) and loss modulus (*G″*) were run within the LVR at an amplitude of 0.1% and a frequency of 1 Hz, where *G′* and *G″* remained invariant and the sample did not undergo structural modifications. 

The determination of the gelation transition temperature (T_sol–gel_) of formulations was performed using the previously described conditions carrying out a temperature sweep analysis over temperatures ranging from 20 to 40 °C and a heating rate of 1 °C/min. 

#### 2.2.4. Measurement of the Adhesive and Mucoadhesive Forces

The adhesive and mucoadhesive behavior of formulations presenting different mucoadhesive agents (Carbopol C971P (carbomer), Protanal LF 10/60 FT (sodium alginate) and Satiaxane (xanthan gum)) were evaluated by measuring the force and the work required to detach the corresponding formulation from the upper plate of the rheometer in a tensile test. This test was carried out with an Anton Paar MCR102 Rheometer and the raw data were analyzed using the Anton PaarRheoCompass software. The geometry chosen to perform all the experiments was a stainless-steel plate/plate (diameter = 50 mm).

For adhesion experiments, the hydrogel was carefully loaded on the lower plate of the rheometer and heated at 37 °C for ensuring the gelation of the system. After that, the upper plate was lowered until contact with the sample (100 µm). After a contact time of 1 min with the hydrogel with a contact force of 0.3 N, the upper plate (rheometer mobile) was moved upward at a constant speed of 5000 µm/s. The strength was recorded as a function of the displacement (elongation), which allowed us to determine the maximal detachment force (F_adh_) and the work of adhesion (W_adh_), calculated by determining the maximal peak and the area under the curve from the force–elongation curve. 

For mucoadhesion experiments, 5% (*w*/*v*) porcine gastric mucin dispersion was previously prepared in sterile water. Mucin films were prepared directly on the upper plate of the rheometer. The mucin dispersion (300 µL) was placed and spread homogeneously on the rheometer mobile and heated during 15 min at 37 °C in the oven allowing the mucin dispersion to dry. Then the hydrogel was loaded on the lower plate and the rheometer mobile was brought to contact under the same experimental conditions as described above for adhesion experiments.

#### 2.2.5. Flow-Through USP-4 Apparatus In Vitro Drug Release

Dexamethasone acetate (DMA) release from P407-HP-β-CD hydrogels was evaluated at 37 °C using an USP-4 apparatus Sotax CE7 Smart (Sotax AG, Nordring, Switzerland) equipped with 22.6 mm diameter cells [71,72]. A ruby bead (5 mm diameter) was placed at the base of the sample cell. Of the hydrogel sample 0.5 g were placed in a Float-A-Lyzer and then inserted in the USP-4 cell. To fill the bottom conical part of the sample cell, 4 g of 1 mm glass beads were added. Of HBSS buffer (pH = 6.8) 50 mL maintained at 37 °C were used as release media in these studies (final concentration of DMA in the release media was 10 µg/mL, corresponding to 100% of drug release). The media was perfused in a closed-loop setting at 8 mL/min. The released DMA amount was monitored every 3 min by UV absorption at 242 nm as the wavelength. An equal amount of DMA complexed with HP-β-CD in solution (10 µg/mL) was placed in the release media directly as a control sample of the complete release of DMA from hydrogel formulations. The cumulative release (%) of DMA from hydrogels was calculated by dividing the DMA amount released from the hydrogel formulation by the measured amount of released DMA from the control solution.

To compare dissolution profiles, the difference factor (ƒ_1_) and similarity factor (ƒ_2_) were determined for each hydrogel formulation release profile, comparing them to the free mucoadhesive dissolution profile [73,74]. The ƒ_1_ and ƒ_2_ factors provide an independent-model method for data analysis to measure the similarity between two release profiles.

The factor ƒ_1_ is the percentage of difference between two dissolution profiles at each time point and is calculated using the following expression:(6)$${ƒ}_{1}=\;\left\{\left[Ʃ\left|R_{t}-T_{t}\right|\right]/ƩR_{t}\right\}\;\times \;100$$
where *R_t_* is the amount of released drug on the reference formulation and *T_t_* is the amount of released drug on the test formulation. If the release profiles are completely superimposed, the *ƒ_1_* factor reaches a value of 0. From a practical point of view, values between 0 ≤ *ƒ_1_* ≤ 15 could be considered as superimposed release profiles.

The factor *ƒ_2_* is calculated using the following expression [75,76]: (7)$${ƒ}_{2}=50\;\times \;\log\left\{{\left[1/1+\left(Ʃ\;{\left(R_{t}-\;T_{t}\right)}^{2}\right)/N\right]}^{-0.5}\;\times \;100\right\}$$
where *N* is the number of experimental data values. If the profiles are completely superimposed, the *ƒ_2_* factor reaches a value of 100. From a practical point of view, values between 50 ≤ *ƒ_2_* ≤ 100 can be considered as superimposed release profiles. 

#### 2.2.6. Statistical Analysis

Statistical analyses were performed using Prism version 7.0 software (GraphPad, Northside, CA, USA). Normality of experiment data was examined using a Shapiro–Wilk test. A one-way analysis of variance test (ANOVA) followed by a Tukey’s multiple comparison post-test comparing means between different groups were performed. The level of statistical significance was set at *p*-value < 0.05. 

## 3. Results and Discussion

### 3.1. Influence of the Hydrogel Preparation Approach on the Physicochemical Behavior of Hydrogels

Both hydrogel preparation approaches were inspired in the called “cold method” consisting of the solubilization of P407 chains at 4 °C for 24 h [36,67]. To improve and facilitate the dispersion of poly(ethylene oxide)-poly(propylene oxide)-poly(ethylene oxide) (PEO-PPO-PEO) polymer chain blocks, the P407 powder was dispersed in a first volume of sterile water then in a second one. In the first approach, this second volume added consisted of an aqueous solution containing complexes of DMA and HP-β-CD molecules but also in a small amount their free-complexed state (Figure 1). The resulting solution was stored for 24 h at 4 °C. At this moment, supramolecular interactions between hydrophobic regions of P407 and HP-β-CD cavities may exist leading to the formation of PPO-HP-β-CD inclusion complexes and the non-complexation of DMA molecules, even relocation of the DMA [77,78,79]. Indeed, when this solution is left at room temperature (15–20 °C), molecules of DMA are precipitated leading to a white coloration of the anterior limpid preparation (DMA is not anymore complexed and not soluble in water in its free state). At these levels of temperature, P407 micelles could not be correctly self-assembled in the presence of HP-β-CD leading to the formation of a very weak hydrogel whereas P407 is shared between CD and other P407 molecules therefore in too small of an amount. 

For the second approach, the P407 dispersion was completed with a second volume of distilled water and stored at 4 °C under magnetic stirring overnight to solubilize all the polymers chains (Figure 2). The following day this solution was placed at room temperature (15–20 °C) in order to obtain the micellization process. After that, DMA-HP-β-CD solution was added and the resulting solution was heated at 30 °C to allow the gelation process. The DMA-HP-β-CD solution was prepared 3 days before. FT-IR and X-ray diffraction patterns reported in the scientific literature demonstrated the formation of DMA-HP-β-CD inclusion complexes [80]. The final product obtained was a strong hydrogel without any DMA precipitation. In this second approach, the prior micelle formation avoided the formation of inclusion complexes between PPO blocks of P407 and HP-β-CD.

### 3.2. Phase Solubility Diagrams and Apparent Solubility Determination

The influence of the presence of HP-β-CD in the solubilization of DMA was investigated by adding an excess of DMA powder in different vials containing increasing concentrations of HP-β-CD and allowing to stir during 72 h to reach the complexation equilibrium. Previous empirical studies have already demonstrated by ^1^H-NMR and ITC that dexamethasone phosphate (DMX) interacts with HP-β-CD molecules to form stable inclusion complexes after 3 days [81,82,83].

The standard curve was prepared in an aqueous solution of DMA-HP-β-CD because of the matrix effect highlighted (data not shown). Indeed, molar extinction coefficient values were different for a propylene glycol or aqueous HP-β-CD standard curve (data not shown). For the standard curve DMA and HP-β-CD were used at concentrations of 1.15 mM and 19.2 mM respectively. These molar concentrations were selected through a prior standard curve in propylene glycol, ensuring the total dissolution of DMA in the HP-β-CD aqueous solution.

Phase solubility diagrams at different HP-β-CD concentrations (Figure 3) were obtained in order to evaluate the apparent solubility of DMA.

The analysis of these diagrams could give an overview of the process of complexation between DMA and cyclodextrin (CD) molecules based on Higuchi and Connors classification (69). Thus, there are two mains groups describing the relationship between the increasing dissolved drug concentration when CD concentration is increased: A-type and B-type. A-type phase-solubility curves exhibit increased drug solubility by increasing CD concentration. The increased solubility can be linear with the increasing host concentration (A_L_-type) or can present negative or positive deviations (A_N_- and A_P_-type) from the linearity explained by the self-association or aggregation of the cyclodextrin molecules or their complexes. For B-type, the phase-solubility diagrams of the formed-complexes show a limited solubility in the aqueous solution or suspensions. Thus, B_S_-type exhibited reduced solubility with a maximal solubility in a concentration plateau. B_I_-type shows insolubility profiles at all the concentrations in the aqueous medium [84].

Higuchi and Connors phase-solubility diagram [69] showed a linear increase in DMA solubility as a function of HP-β-CD concentration, suggesting the formation of 1:1 inclusion complexes between HP-β-CD and DMA (A_L_-type) [85,86]. Without any further additional information, it is usually made that the assumption of 1:1 complex stoichiometry meaning that one molecule of HP-β-CD is complexing one molecule of DMA. However, the A_L_ diagram means that only one molecule of HP-β-CD is involved in the interaction with one or more DMA molecules. These results are in agreement with other empirical studies. Indeed, ITC titration profiles reported in the scientific literature showing exothermic heats highlighting the interaction between β-CD molecules and DMA. The corrected heat of reaction per mole curves showed also a very good fit with a 1:1 binding stoichiometry model [82,87]. In addition, the same study reported the Job’s plots showing the intersection of two lines at a mole fraction of 0.52, which corresponds to a 1:0.92 binding stoichiometry [82,87].

A more quantitative description of the complexation behavior can be achieved by considering the value of the binding constant (also called stability constant or complex equilibrium constant) calculated according to the Equation (8) and assuming a 1:1 interaction stoichiometry: (8)$$k_{1:1}=\;\frac{slope}{S_{0}\;\left(1-slope\right)}$$
where *k_1:1_* is the calculated constant from the slope of the phase-solubility diagram, *S_0_* is the intrinsic solubility of the drug in the absence of HP-β-CD molecules (this value should be equivalent to the *S_int_*, which corresponds to the y-intercept value of the phase solubility plot in the linear region). In our study the value of *k_1:1_* was 3124 M^−1^, a little higher than in the literature [66,87,88,89]. This value of stability constant is very high and corroborates that, once the inclusion complexes are formed, they are relatively stable, confirming that the second method of preparation of hydrogels is the most suitable to obtain stable formulations. Values of *k_1:1_* between 200 and 5000 M^−1^ are considered as the most suitable to improve the bioavailability of poorly water soluble drugs [80,90].

The complexation of DMA in HP-β-CD led to the increase of the apparent solubility of DMA. The presence of HP-β-CD molecules allowed it to reach a DMA concentration of 1.6 mM, representing an apparent solubility enhancement factor of 67 (*S*_0_ = 0.024 mM [91,92]. The complexation efficiency (CE) was also calculated from Equation (9), giving a value of 0.075.
(9)$$CE=\;S_{0}k_{1:1}=\;\frac{\left[DMA\right]\left[HP-\beta -CD\right]}{HP-\beta -CD}=\;\frac{slope}{\left(1-slope\right)}$$

The main goal of phase-solubility diagrams was to investigate the interaction stoichiometry and the HP-β-CD concentration necessary to reach a DMA apparent solubility of 2.30 mM (0.10% *w*/*v*). According to the phase-solubility plot, the HP-β-CD threshold concentration to reach it was 33.6 mM (4.54% *w*/*w*). This HP-β-CD concentration was kept for the next experiments.

### 3.3. Rheological Studies

The ideal candidate for the controlled delivery of an API to the buccal cavity should exhibit a wide variety of characteristics mainly including the ease of application, the low elimination and the high retention time into the cavity once the drug is administered. For these reasons, rheological characterization seems to be essential to select the ideal product profile. Table 2 and Table 3 show the compositions and the main rheological parameters obtained for the different formulations.

#### 3.3.1. Oscillatory Rheometry Studies

Oscillatory studies were performed in order to determine *G′*, *G″* and tan δ in the LVR of different formulations. Concerning the two dynamic moduli acquired, the higher the *G′* value is, the more expanded the elastic behavior is. In the other hand and conversely, the higher the *G″* is, the more pronounced the viscous behavior is. We can usually consider that one formulation exhibits a viscous-dominant behavior (liquid-like) when the *G″* value is higher than the *G′* value. Contrarily, when the *G′* value is higher than the *G″* value, the formulation exhibits an elastic-dominant behavior (solid-like) at a given oscillatory frequency. The *G″*-to-*G′* ratio gives tan δ, which is a measure of the relative contribution of the viscous behavior to the mechanical properties of materials. A gel (solid-like) state is observed when tan δ ≤ 1 [36,93].

*G′* and *G″* dynamic moduli were used to determine the T_sol–gel_ of formulations. Indeed, the temperature corresponding to the crossover of the elastic and viscous moduli was considered as the gelation transition temperature (Figure 4).

The possible effect of HP-β-CD molecules and the hydration ratio of P407 polymer chains were investigated by comparing the viscoelastic properties and T_sol–gel_ of formulations A, B and C (Table 2). Comparative analysis between formulations A and B (equal mass ratio) and A and C (equal hydration ratio) highlighted the significant increase of gelation transition temperature when HP-β-CD was added (*p*-value = 0.0008 and <0.0001 for the comparison between formulation A and formulation B and C respectively after a one-way ANOVA test). Indeed, the flow heat provided to form the gel needs to be more important than previously described, because HP-β-CD molecules are located between P407 polymer chains, leading to a steric blockage. In one hand, the heat supplied to self-assemble P407 polymer chains in micelles may be higher, leading to an increase of the critical micellization temperature (CMT). In the other hand, once the micelles of P407 are well-formed, the HP-β-CD hindered molecules decrease the micelle interactions between themselves [94,95]. That will result in an increase of the crystallization temperature. Comparative analysis of formulations B and C (equal HP-β-CD but different hydration ratio) confirmed that the hydration ratio of P407 polymer chains plays an essential role in the sol–gel transition. Indeed, the higher the hydration ratio is, the higher the T_sol–gel_ will be. The formulation C, showing a hydration ratio more important than the formulation B, exhibited a higher T_sol–gel_. The heat provided to induce the gelation need to be higher because water molecules in the micellar core are increased and the micellization is disfavored, leading to an increase in the CMT and subsequently in the T_sol–gel_ [93].

The influence of the DMA addition in the viscoelastic properties of hydrogels was evaluated by comparing the T_sol–gel_, *G′* and *G″* of formulations D and E with a control sample without active substance (formulation C), as reported in Table 2. Since the water-soluble complexes formed between HP-β-CD and DMA are not expected to enter into the hydrophobic region of poloxamer micelles, the hypothesis is that the inclusion complexes would be located in the aqueous region of hydrogels. Indeed, when cyclodextrin molecules are added on the complexed-state with DMA molecules (formulation E), those ones may interact less with hydrophobic blocks of PPO P407 chains, favoring the micellization and crystallization of P407 polymer chains. That may reduce the T_sol–gel_, as we can observe the results shown in Table 2. However, when CD molecules are added in their free-complexed state (formulation C), they may establish more interactions with hydrophobic poloxamer block chains, leading to an increase on the transition temperature (*p*-value = 0.0064 for the comparison between formulations C and E after a one-way ANOVA test).

Hydration ratio of P407 was firstly identified as a key parameter controlling the viscoelastic properties of formulations. In order to retrieve an equivalent T_sol–gel_, the hydration ratio of formulation E was decreased (formulation D), and the initial transition temperature without active substance was recovered (*p*-value = 0.0011 for the comparison between formulation D and E after a one-way ANOVA test). Thus, the formulation D, containing a hydration ratio of 0.191 was kept for experiments in the presence of mucoadhesive agents.

The effect of mucoadhesive agents on the viscoelastic properties of P407-based hydrogels was also investigated (Table 3). The transition temperature did not significantly change whatever the mucoadhesive agents used (*p*-value > 0.05 for the comparison between all the formulations containing mucoadhesive agents (samples F-K) and the control sample (sample D) after a one-way ANOVA test). All the mucoadhesive hydrogels showed T_sol–gel_ above 29.5 °C and very close to the control sample (formulation D, without mucoadhesive agent). However, regarding the elastic modulus values at 37 °C, we could find significantly lower values than the control sample, suggesting the unfavorable interaction between the mucoadhesive agent and P407-hydrogel (*p*-value < 0.0001). The mucoadhesive properties of hydrogels containing a mucoadhesive agent need to be investigated to justify the addition of those products in the formulations [96].

#### 3.3.2. Flow Rheometry Studies

Viscosity and flow properties were performed at 25 and 37 °C and were shown in Figure 5. These properties may determine the ease of administration of the product into the buccal cavity (generally at room temperature) by spraying and the time-dependent clearance of the formulation after the administration (at physiological temperature). 

In both cases, 25 and 37 °C, all the formulations, except the control sample without mucoadhesive agent (sample D) and 0.1 wt% alginate hydrogel (sample H) at 25 °C, exhibited a shear-thinning behavior. Viscograms were fitted with a Carreau-Yasuda model. Correlation coefficient values were satisfactory for all the formulations. This model describes the shear-thinning behavior of polymer solutions or suspensions presenting two Newtonian plateaus for very low and very high shear rates, representing the low and high deformation states of hydrogels respectively.

At 25 °C, the major goal of this study was to evaluate if the hydrogel formulations could be applied by spraying. A pharmaceutical solution for spraying may exhibit a low viscosity at the steady state, presenting usually values η < 200 mPa-s at a shear rate around 340 s^−1^. Thus, as we could observe in Figure 5, at 25 °C, all the mucoadhesive hydrogel formulations could be used for spraying applications.

The goal of flow rheometry studies at 37 °C was to investigate the texture and the hold of the hydrogel formulations once the T_sol–gel_ was reached. The shear stress developed and associated to a stimulus for viscous liquids and foods seems to be constant and corresponds to shear rate values about 10 s^−1^ [29,97]. As observed in Figure 5, all the hydrogel formulations containing a mucoadhesive agent (samples F-K) exhibited, at 10 s^−1^ and 37 °C, viscosity values lower than the control value (sample D without mucoadhesive agent). These results, and according to viscoelastic properties previously described, suggested a partial loss on the crystalline structure of P407 micelles when mucoadhesive matrices are incorporated in the hydrogel. The formulations showing the closest behavior to P407-based hydrogels without mucoadhesive agent at 37 °C were the preparations containing the xanthan gum (at 0.05 and 0.1 wt%, samples J and K respectively) and sodium alginate (at 0.5 wt%, sample I).

### 3.4. Measurement of the Adhesive and Mucoadhesive Forces

Mucoadhesion is considered as a phenomenon creating an intimate contact between two surfaces, at least one of them being biological, remaining together for a prolonged period of time. In this context, it involves the intimate interaction between the dosage form and the mucosa. This contact is believed to be the result of the wetting and adsorption of hydrogels on the mucosa leading to the diffusion and interpenetration of hydrogel polymer chains with mucosal compounds. Therefore, polymer chains could establish supramolecular interactions with mucosa such as hydrogen bonds, electrostatic interaction or hydrophobic contacts.

The experimental method used and developed in this publication was inspired and modified from a previously described method by Ponchel [98,99]. This method is very useful, because more information can be reached in order to obtain different deformation parameters.

In a typical tensile experiment, the hydrogel was placed between the lower plate and the upper plate, covered or not with porcine gastric mucins. In these experiments, the force necessary to produce the detachment of the two surfaces was recorded as a function of the elongation distance between the upper surface and hydrogel formulation. In a typical force–elongation curve, several distinguished regions could be observed in Figure 6 [93,99]. 

The first part of the force–elongation curve (segment AB) corresponds to the force increase as a function of the elongation distance to reach the peak, corresponding to the maximal force required to detach the two surfaces (F_max_). In this position of the curve, the contact surface between the hydrogel and the mucins, adsorbed on the upper plate (rheometer mobile), remained constant and is equal to the contact surface of the hydrogel. The second part of the curve corresponds to the segment BC, which will be the partial detachment period of the rheometer geometry from the hydrogel formulation. A slight decrease of the contact area is occurring. The final step of the curve (segment CD) indicates the total detachment of the mucin mobile from the lower support where the major change of the contact surface is occurring. Finally, the W part of the curve will correspond to the area under the curve leading to the adhesion work value (W_adh_). The analysis of the whole curve and not only the peak of detachment could give us some information about adhesive interactions between both materials. In this way, large deformation peaks would mean strong adhesive and more compliant interactions.

Comparing deformation parameters (F_max_ and W) in the presence and in absence of mucin, we observed a significant increase for those performed with mucins (Table 4). That indicates a positive interaction between hydrogels and mucins, which is coherent with data reported in the literature [46,48,50].

From the results reported in Figure 7, adhesion and mucoadhesion performed experiments confirmed that mucoadhesive agents included in the three-dimensional network of hydrogels were useful to increase the adhesive behavior of formulations, excepting the use of Carbopol (sample F).

The control sample, prepared from a mucoadhesive agent-free P407-based hydrogel (sample D), exhibited more resistant deformation parameters than Carbopol C971P hydrogel preparations (sample F, *p*-value > 0.05 after a one-way ANOVA test). Nevertheless, we can observe that the maximal forces required to detach the upper plate from the mucoadhesive hydrogel containing sodium alginate (samples H and I) or xanthan gum (samples J and K) were higher than the control sample (*p*-value < 0.05 for sodium alginate and xanthan gum hydrogel formulations). Despite maximal forces of detachment for xanthan gum being smaller than sodium alginate hydrogels, xanthan gum formulations seem to be slightly more mucoadhesive than sodium alginate regarding the work of adhesion. The higher elasticity (*G′* values at 37 °C) of P407-based hydrogels containing xanthan gum (samples J and K) could explain this difference, because the possibility of micelles to interact with mucin compounds would be increased. Furthermore, the increased viscosity at 37 °C of xanthan gum formulations, compared to sodium alginate (samples H and I) and Carbopol C971P (sample F), would increase the contact time between mucin and mucoadhesive agent, leading to a stronger mucoadhesive behavior.

### 3.5. Flow-Through USP-4 Apparatus In Vitro Drug Release

The main goal of this study was to establish an in vitro release test with mucoadhesive hydrogels allowing it to release 70–80% of the drug amount from the hydrogels within 30 min. This requirement comes from clinical demands to improve observance and efficacy of drug treatment. The in vitro release assay was performed in a liquid media mimicking physiological conditions in the oral cavity in terms of pH, salivary clearance and mechanical stress.

In this context, a simple exponential expression [100,101], as follows in the Equation (10), could be useful to analyze the controlled-release behavior of our hydrogels in the presence of xanthan gum at two different concentrations.
(10)$$\frac{M_{t}}{M_{\infty }}=k\;\times t^{n}$$
where, *M_t_/M_∞_* is the fractions of drug released (*M_t_* is defined as the amount of drug released at time *t* and *M_∞_* is defined as the amount of drug released at time infinity), *k* is the kinetic constant, *t* is the release time and *n* is the diffusional exponent for drug release.

The diffusion exponent of this simple equation may give us some important information about the drug release mechanism [102]. In this context, and for a cylindrical matrix, if the exponent *n* ≤ 0.45, then the drug release mechanism is the Fickian diffusion. Fickian diffusion involves the molecular diffusion of the drug due to a chemical potential gradient. If *n* ≥ 0.89, then the drug is released by a case II transport, also called relaxational transport or erosion (typical zero order release). In this case, this phenomenon involves the penetration of the dissolution media into the polymer matrix at a controlled-rate modulating the drug release. Both phenomena are considered as the limiting case [71].

Many release processes, especially from hydrogels and swellable polymer matrices, are placed between the two limiting cases previously described: the simple diffusion and the erosion transport mechanism. In the case of both phenomena are coexisting, this case involves a diffusion exponent 0.45 < *n* < 0.89 where the prevailing mechanism would a combination of simple diffusion and macromolecular relaxation because of the surrounding medium. For this reason, another equation may describe their drug release behavior from a thin polymer film, as follows: (11)$$\frac{M_{t}}{M_{\infty }}=\;k_{1}t^{0.5}+\;k_{2}\;t$$
(12)$$\frac{M_{t}}{M_{\infty }}=4\;{\left(\frac{D\;t}{\pi \;t^{2}}\right)}^{0.5}=\;k_{1}t^{0.5}$$
(13)$$\frac{M_{t}}{M_{\infty }}=\;\frac{2\;k_{0}}{C_{0}\;l\;}\;t=k_{2}\;t$$

The first term of the Equation (11) corresponds to the Fickian contribution, where *k_1_* is the diffusion constant and represents the short-time approximation of the fractional drug released (Equation (12)) where *D* is the drug diffusion coefficient, *l* is the initial film thickness and *t* is the release time. The second term of the Equation (11) represents the relaxational contribution or erosion, where *k_2_* is the relaxation constant and defined as previously described in Equation (13) where *k_o_* is the relaxation constant and *C_o_* is the drug released concentration [100,101,102].

For further analysis of our data, a first fit of Equation (10) (expressing the drug release data versus time) was made for *M_t_/M_∞_* up to 100% of the release as shown in Figure 8. As we can observe, DMA drug release from a P407-based hydrogel in the absence of a mucoadhesive agent was extremely high after a few minutes. Our strategy was to incorporate mucoadhesive polymers such as Carbopol C971P (sample F), sodium alginate (samples H and I) or xanthan gum (samples J and K) in order to slow down the release of DMA. As we can see in Figure 8, Carbopol C971P and sodium alginate showed release profiles very close to the mucoadhesive-free P407-based hydrogel (sample D). Regression parameters from the Korsmeyer–Peppas model showed a good fit (Table 5), meaning that the DMA release is mainly due to diffusion and in an immediate manner. The total amount of DMA was released over 10 min. 

However, one mucoadhesive compound was revealed as a promising agent, the xanthan gum (samples J and K). It is widely described in the scientific literature that xanthan gum can be used as an effective mucoadhesive agent to form a polymeric matrix enabling the extension of drug release [103]. An inverse relationship between the amount of xanthan gum present in the formulation and the drug release rate has been found [103,104]. In this case, the drug may be released at a constant rate pH-independent in a zero-order release guarantying a constant rate of drug release versus time [105]. The regression parameters of the Korsmeyer–Peppas fitted model were not good for released amounts of DMA up to 100%. Nevertheless, if fitted up to 60% of the total released amount of DMA (short time release), the correlation model was very high. At short times, up to 60% of total drug released, the *n* values are comprised between 0.45 and 0.89 indicating that two limiting phenomena, Fickian diffusion and erosion, are occurring in the hydrogel. A lower value of *n* for the higher concentration of xanthan gum indicates that a diffusion process is more predominant than for the lower concentration.

Regarding the second fitted-model data in Table 6, for short-times, we observed that for both formulations diffusion constants are higher than the relaxation contribution. When comparing both concentrations, we noted that diffusion is more important in 0.1% xanthan gum-based hydrogels (sample K) than formulations containing a lower concentration, sample J [103]. As well, for relaxation contribution, it is stronger for the 0.05% xanthan gum-based hydrogel. 

Thus, xanthan gum may be an important compound controlling the drug release by diffusion phenomena and decreasing the relaxation of the hydrogel at short-time. If the experimental data were fitted to long-times, we realized that erosion mechanism did not exist anymore and the only phenomenon occurring was the Fickian diffusion.

A comparison of the release profiles of mucoadhesive-free P407-based hydrogel and those containing mucoadhesive compounds was performed with an independent-model analysis [106]. The *ƒ_1_* (difference) and *ƒ_2_* (similarity) factors were comprised between 0 ≤ *ƒ_1_* ≤ 15 and 50 ≤ *ƒ_2_* ≤ 100 respectively for all hydrogel formulations except those having xanthan gum as the mucoadhesive matrix. In that case, difference and similarity factor values were *ƒ_1_*> 15 and *ƒ_2_*< 50 respectively indicating that their dissolution profiles were very different from the control sample. That confirms the effect of xanthan gum on the controlled-release profiles of P407-based hydrogel as previously suggested. 

As previously described in the scientific literature, the surfaces of hydrogels tend to form a swollen layer to prevent a rapid penetration of the dissolution media, which may inhibit or limit the erosion of the formulation. In the other hand, sodium alginate, used as a controlled-release excipient (samples H and I) showed faster dissolution rates as previously described [107]. The potential presence of alginic acid molecules, due to slight variations on the pH, may act as a barrier for the media penetration. Those non-swellable properties could induce a strong erosion or dissolution of alginate-based hydrogels [104,107].

Based on Peppas–Sahlin fitted model (Figure 9) at short and long-times, the times for 25, 50, 80 and 90% of total released amount were calculated and the average values were shown in Table 7.

## 4. Conclusions

Self-assembly of poloxamer 407 chains forming thermoresponsive hydrogels is still a promising approach to build new therapeutic mucosal and topical dosage forms using soft conditions without solvent, surfactants or any pH modifications. These hydrogels have been very useful for the treatment of topical or mucosal diseases by incorporating a therapeutic drug into the three dimensional network. However, its use is quite limited because of the lack of mucoadhesive properties. In this work, and in the context of the treatment of oral planus lichen, a chronic dermato-mucosal disease, we developed poloxamer P407-based hydrogels containing a poorly water soluble anti-inflammatory drug encapsulated into the cavity of HP-β-CD, dexamethasone acetate (DMA), with improved mucoadhesive behavior. Indeed, including mucoadhesive agents such as xanthan gum or sodium alginate increased significantly the adhesive properties of hydrogels. The effect of experimental formulations parameters, such as the poloxamer hydration ratio, presence of HP-β-CD and its state and the incorporation of mucoadhesive agents, on the rheological behavior was evaluated to identify key parameters.

Furthermore, in vitro drug release studies were performed. One of the mucoadhesive agents was identified as an in vitro controlling release compound. Xanthan gum, incorporated in the hydrogel formulation, was able to sustain drug release for longer times than the other compounds and the mucoadhesive-free hydrogel. This work opens the way to a new approach easing the design of mucoadhesive sprayable hydrogels for the treatment of mucosal diseases with water-insoluble drugs.

## Figures and Tables

**Figure 1 pharmaceutics-13-00117-f001:**
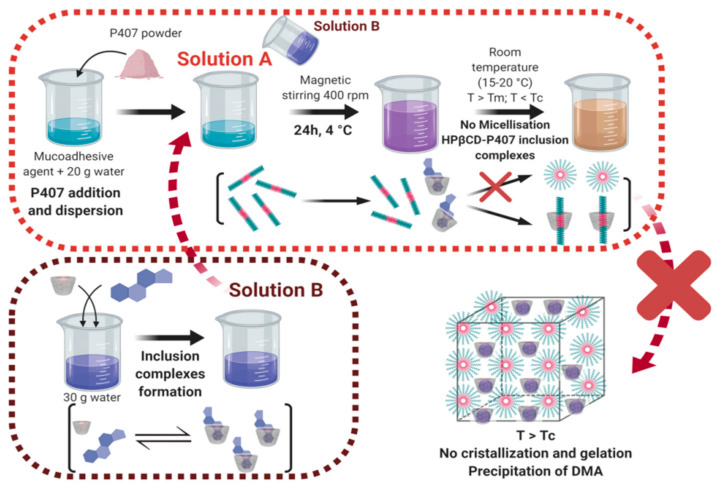
Schematic representation of the association of poloxamer P407 with HP-β-CD dethreading the inclusion complexes previously formed with DMA in sterile water.

**Figure 2 pharmaceutics-13-00117-f002:**
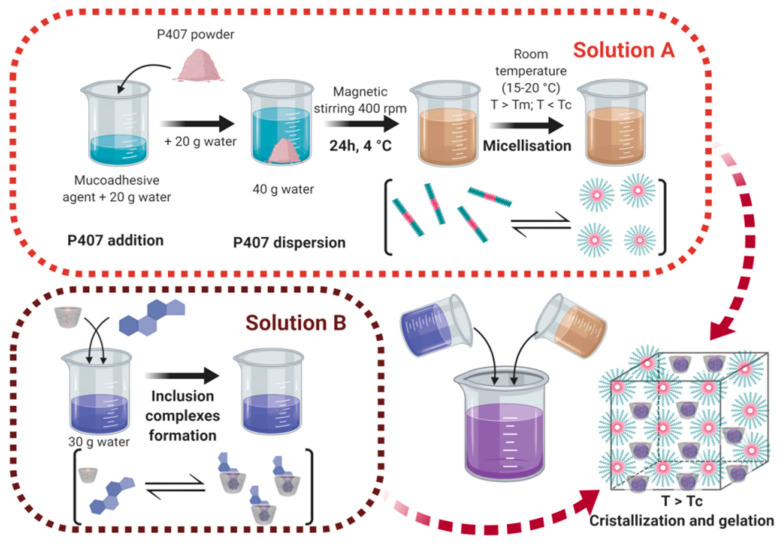
Schematic representation of the self-association of poloxamer 407 and complexation of HP-β-CD with DMA in sterile water to form poloxamer P407-based hydrogels.

**Figure 3 pharmaceutics-13-00117-f003:**
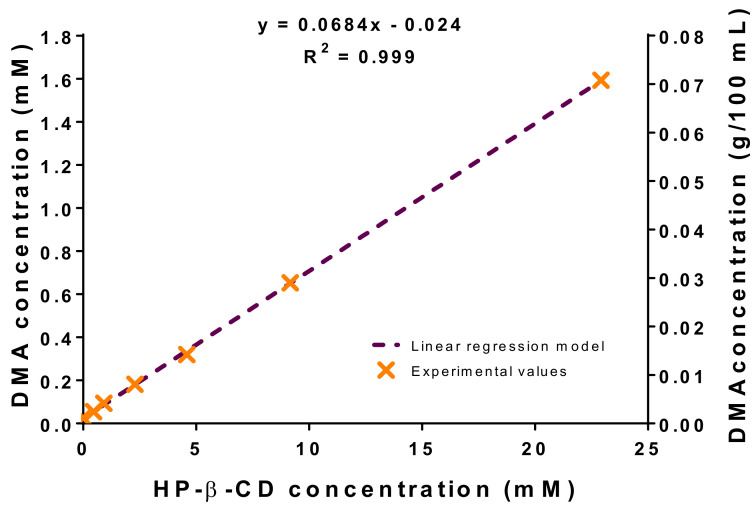
Experimental phase solubility diagram of DMA for HP-β-CD (*n* = 3).

**Figure 4 pharmaceutics-13-00117-f004:**
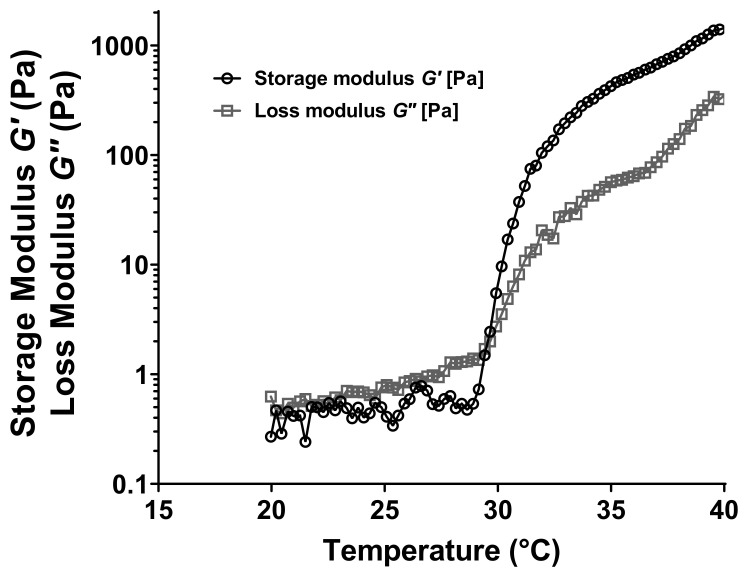
Typical profile of the variation of the storage (*G′*) and loss (*G″*) moduli as a function of temperature. Experiments performed with a poloxamer P407-based hydrogel at 16% (*w*/*w*) containing xanthan gum at a concentration of 0.1%, sample K (*w*/*w*).

**Figure 5 pharmaceutics-13-00117-f005:**
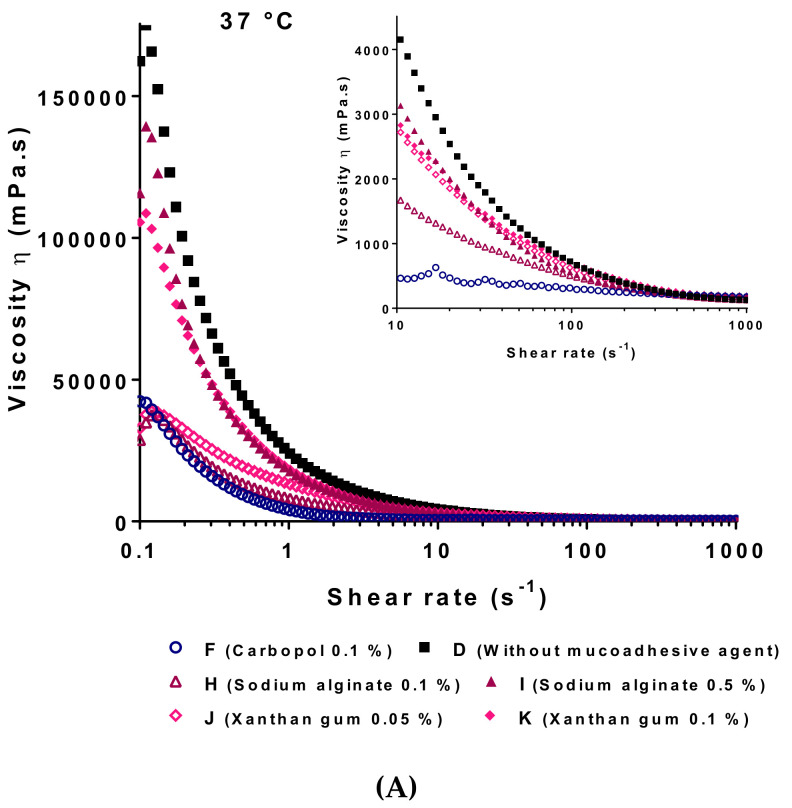
Dynamic viscosity as a function of shear rate at 37 (**A**) and 25 °C (**B**) of various poloxamer P407-based hydrogels. The mean of 3 experiments are represented.

**Figure 6 pharmaceutics-13-00117-f006:**
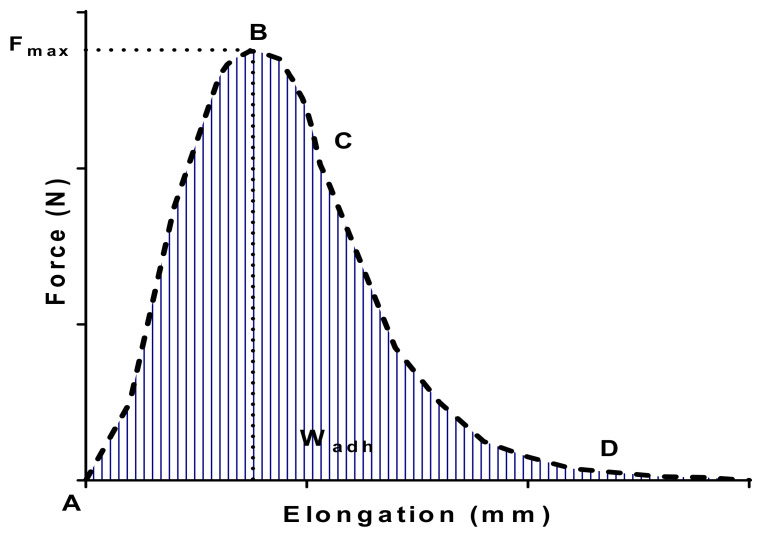
Typical force versus elongation curve for poloxamer P407-based hydrogels. The analysis of the curve can lead to the determination of the maximal force of detachment (F_max_) and the work of adhesion (W_adh_).

**Figure 7 pharmaceutics-13-00117-f007:**
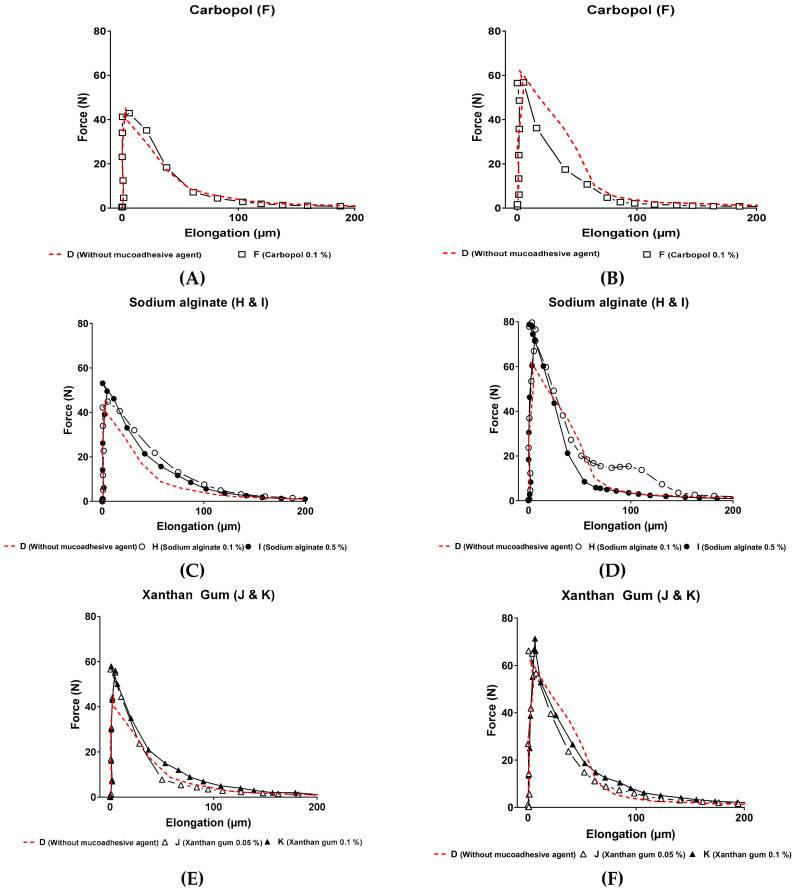
Forces curves as a function of elongation for Carbopol C971P sample F (**A**,**B**), sodium alginate samples H and I (**C**,**D**) and xanthan gum samples J and K (**E**,**F**). Force–elongation curves in the left side (**A**,**C**,**E**) correspond to adhesion profiles (without mucin). Curves in the right side correspond to mucoadhesion profiles (with mucin) (*n* = 3).

**Figure 8 pharmaceutics-13-00117-f008:**
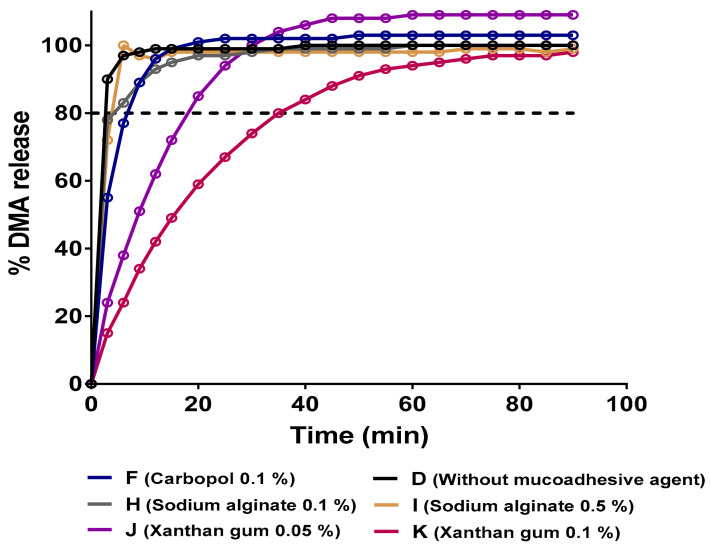
In vitro DMA (dexamethasone acetate) profiles from various hydrogels using the USP4 flow-through apparatus at 37 °C. The mean values of 3 experiments are represented.

**Figure 9 pharmaceutics-13-00117-f009:**
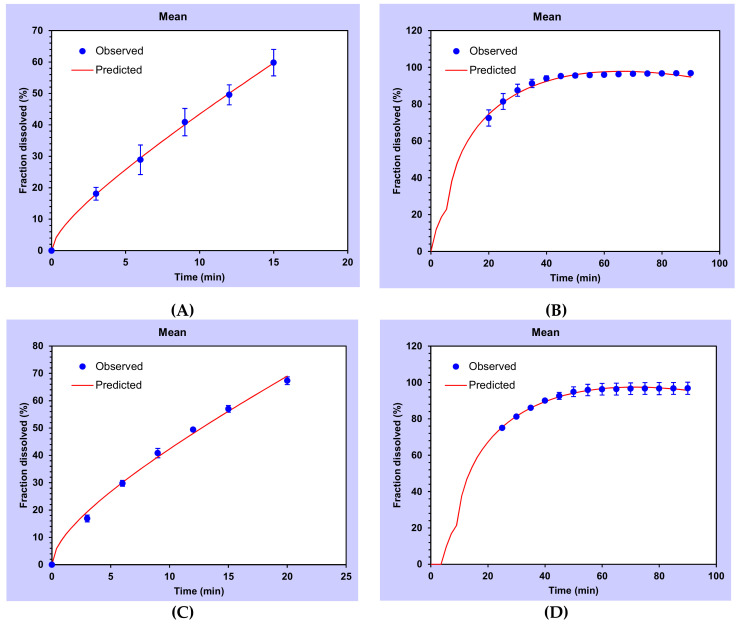
Drug release data fitted to a Peppas–Sahlin dissolution model. Short-times fit (**A**–**C**) and long-times fit (**B**–**D**) for poloxamer 407-based hydrogels containing xanthan gum at 0.05 (**A**–**B**) and 0.10 (**C**–**D**) wt% as a mucoadhesive agent.

**Table 1 pharmaceutics-13-00117-t001:** Qualitative and quantitative composition of hydrogels.

	P407 (wt%)	P407/H_2_O Ratio	H_2_O (wt%)	HP-β-CD (wt%)	DMA (wt%)	MA Agent	MA (wt%)	Use
A	17.00	83.00	0.205	0	-	-	-	Physicochemical characterization
B	17.00	78.13	0.218	4.87	-	-	-	Physicochemical characterization
C	16.17	78.96	0.205	4.87	-	-	-	Physicochemical characterization
D	15.21	79.82	0.191	4.87	0.10	-	-	Physicochemical characterization
E	16.16	78.87	0.205	4.87	0.10	-	-	Physicochemical characterization
F	15.21	79.82	0.191	4.87	0.10	Carbopol 971P	0.10	MA in vitro tests and in vitro dissolution profile
G	15.21	79.82	0.191	4.87	0.10	Carbopol 971P	0.50	MA in vitro tests and in vitro dissolution profile
H	15.21	79.82	0.191	4.87	0.10	Sodium alginate	0.10	MA in vitro tests and in vitro dissolution profile
I	15.21	79.82	0.191	4.87	0.10	Sodium alginate	0.50	MA in vitro tests and in vitro dissolution profile
J	15.21	79.82	0.191	4.87	0.10	Xanthan Gum	0.05	MA in vitro tests and in vitro dissolution profile
K	15.21	79.82	0.191	4.87	0.10	Xanthan Gum	0.10	MA in vitro tests and in vitro dissolution profile

**Table 2 pharmaceutics-13-00117-t002:** Composition of poloxamer P407-based hydrogels in the presence and absence of HP-β-CD and DMA. Values of the storage (*G′*) and loss (*G″*) moduli, tan δ at 20 and 37 °C, viscoelastic behavior before and after gelation and T_sol–gel_ for poloxomer P407-based hydrogels without mucoadhesive agents. Data were reported as the mean ± SD (*n* = 3).

		Without DMA			With DMA	
		A	B	C	D	E
HP-β-CD (wt%)		0	4.87	4.87	4.87	4.87
P407 (wt%)		17	17	16.17	15.21	16.16
H_2_O (wt%)		83.00	78.13	78.96	79.82	78.87
P407/H_2_O ratio		0.205	0.218	0.205	0.191	0.205
DMA (wt%)		Ø	Ø	Ø	0.1	0.1
20 °C	*G′* (Pa)	0.07 ± 0.08	0.35 ± 0.56	0.02 ± 0.03	0.08 ± 0.08	0.05 ± 0.08
	*G″* (Pa)	0.23 ± 0.03	0.26 ± 0.01	0.20 ± 0.01	0.17 ± 0.06	0.31 ± 0.14
37 °C	*G′* (Pa)	12139 ± 329	10540 ± 291	8052 ± 566	3787 ± 960	9904 ± 641
	*G″* (Pa)	1419 ± 987	772 ± 282	906 ± 27	924 ± 111	1001 ± 44
tan δ at 20 °C		0.32 ± 0.45	1.31 ± 2.06	0.08 ± 0.14	0.36 ± 0.38	0.21 ± 0.37
tan δ at 37 °C		11.47 ± 6.58	14.79 ± 4.69	8.89 ± 0.63	4.05 ± 0.60	9.89 ± 0.45
T_sol–gel_ (°C)		26.4 ± 0.5	28.8 ± 0.0	29.7 ± 0.6	30.2 ± 0.4	27.9 ± 0.6

**Table 3 pharmaceutics-13-00117-t003:** Values of the storage (*G′*) and loss (*G″*) moduli, tan δ at 20 and 37 °C, viscoelastic behavior before and after gelation and T_sol–gel_ for poloxomer P407-based hydrogels containing various mucoadhesive agents. Data were reported as the mean ± SD (*n* = 3).

		Without Mucoadhesive	Mucoadhesive Agents					
		D	F	G	H	I	J	K
		Control sample	Carbopol 971P		Sodium alginate		Xanthan Gum	
			0.1%	0.5%	0.1%	0.5%	0.05%	0.1%
20 °C	*G′* (Pa)	0.02 ± 0.02	0.03 ± 0.03	-	0.08 ± 0.07	0.09 ± 0.17	0.29 ± 0.02	0.61 ± 0.09
	*G″* (Pa)	0.25 ± 0.14	0.33 ± 0.04	-	0.28 ± 0.07	0.69 ± 0.18	0.45 ± 0.16	0.87 ± 0.46
37 °C	*G′* (Pa)	3810 ± 385	1204 ± 14	-	1123 ± 132	1198 ± 107	1528 ± 99	1209 ± 233
	*G″* (Pa)	1073 ± 51	169 ± 69	-	150 ± 6	220 ± 36	369 ± 44	215 ± 79
tan δ 20 °C		13.92 ± 3.85	10.62 ± 3.60	-	3.62 ± 0.12	7.13 ± 0.18	1.60 ± 0.66	1.43 ± 0.70
tan δ 37 °C		0.28 ± 0.02	0.14 ± 0.06	-	0.13 ± 0.01	0.18 ± 0.04	0.24 ± 0.02	0.17 ± 0.03
T_sol–gel_ (°C)		30.5 ± 0.4	30.8 ± 0.6	-	30.2 ± 0.3	31.0 ± 0.2	29.7 ± 0.6	29.6 ± 0.5

ND: not determined, sample G exhibited an elastic behavior (solid-like) at room temperature, without a mucoadhesive agent.

**Table 4 pharmaceutics-13-00117-t004:** Maximal force of detachment (F_max_) and work of adhesion (W) values obtained from adhesion (adh) and mucoadhesion (m-adh) experiments at 37 °C using poloxamer P407-based hydrogels containing various mucoadhesive agents. The control sample consists of a solution of P407 in absence of mucoadhesive agent. ND: not determined, sample G exhibited. The values represented are the mean ± SD (*n* = 3).

Sample	Control Sample (D)	F	G	H	I	J	K
Mucoadhesive Agent	- ^b^	Carbopol C971P		Sodium Alginate		Xanthan Gum	
Wt%	0	0.1	0.5	0.1	0.5	0.05	0.1
F_adh_ (N)	45.5 ± 1.2	42.9 ± 2.9	ND ^a^	44.9 ± 2.8	53.1 ± 1.1	56.6 ± 6.5	58.2 ± 0.6
F_m-adh_ (N)	60.9 ± 1.5	56.5 ± 2.0	ND ^a^	79.7 ± 9.0	78.8 ± 5.9	66.2 ± 3.4	71.4 ± 2.9
W_adh_ (mJ)	2.2 ± 0.1	2.4 ± 0.2	ND ^a^	3.4 ± 0.3	3.1 ± 0.5	2.5 ± 0.2	3.3 ± 0.2
W_m-adh_ (mJ)	3.7 ± 0.3	3.2 ± 0.4	ND ^a^	4.4 ± 0.6	3.6 ± 0.5	3.9 ± 0.3	4.6 ± 0.6

^a^ ND: not determined, sample G exhibited an elastic behavior (solid-like) at room temperature. ^b^ without a mucoadhesive agent.

**Table 5 pharmaceutics-13-00117-t005:** Mathematical model fitting of drug release data from formulations containing various mucoadhesive agents using the Korsmeyer–Peppas model (*n* = 3).

Sample			Korsmeyer–Peppas Fitted-Model			Short-Time Korsmeyer–Peppas Fitted-Model		
Code	MA Agent	[C]	R^2^ Adjusted	n	K	R^2^ Adjusted	n	K
Control	Ø	-	0.9953	0.015	93.5	ND ^a^	ND ^a^	ND ^a^
F	Carbopol	0.1	0.9076	0.094	69.1	ND ^a^	ND ^a^	ND ^a^
H	Alginate	0.1	0.9818	0.055	78.8	ND ^a^	ND ^a^	ND ^a^
I	Alginate	0.5	0.9414	0.043	83.2	ND ^a^	ND ^a^	ND ^a^
J	XG	0.05	*	*	*	0.999	0.76	7.6
K	XG	0.10	*	*	*	0.999	0.69	8.8

^a^ ND: not determined, * Data not shown because of lack of fit for samples J and K.

**Table 6 pharmaceutics-13-00117-t006:** Mathematical model fitting of drug release data from formulations containing xanthan gum as mucoadhesive agent using Peppas–Sahlin model (*n* = 3).

Code	MA Agent	[C]	Peppas-Sahlin Fitted Model					
			Short-Time			Long-Time		
			K_1_	K_2_	R^2^ Adjusted	K_1_	K_2_	R^2^ Adjusted
J	XG	0.05	6.3	2.4	0.998	24.8	-	0.965
K	XG	0.10	8.5	1.6	0.994	24.3	-	0.992

**Table 7 pharmaceutics-13-00117-t007:** Times for 25, 50, 80 and 90% of drug release estimated from short- and long-times Peppas–Sahlin model for hydrogels containing xanthan gum as a mucoadhesive agent (*n* = 3).

Sample			Short-Times Drug Release Time (min)				Long-Times Drug Release Time (min)			
Code	MA Agent	[C] (wt%)	T_25_	T_50_	T_80_	T_90_	T_25_	T_50_	T_80_	T_90_
J	XG	0.05	4.8	12.0	21.5	24.9	4.6	9.1	24.0	35.5
K	XG	0.10	4.5	12.7.	24.6	28.8	8.3	13.0	28.6	41.2

## Data Availability

Not applicable.

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
