# Peer review of "Mucoadhesive Poloxamer-Based Hydrogels for the Release of HP-?-CD-Complexed Dexamethasone in the Treatment of Buccal Diseases"

_pharmaceutics, 2021, doi:10.3390/pharmaceutics13010117_

Round 1
Reviewer 1 Report
This paper covers the development of Poloxamer-based hydrogels for the release of HP-β-CD complexed dexamethasone in the treatment of buccal diseases. Prepared hydrogels were characterized physicochemically and evaluated for in vitro drug release profile. I recommend for publication of this paper after considering the following points:
- Authors have well prepared and characterized the hydrogels. However, in vivo studies are missing in the article. The inclusion of in vivo data would enhance the validity of the developed formulations.
- Abstract: The quantitative information is completely missing in abstract. Kindly included some quantitative information to enhance the readability of the manuscript.
- Introduction: In general, the introduction section is too lengthy. Authors are advised to remove general and well known information from this section.
- The symbols in all equations must be in italics, while their subscripts and superscripts should not be in italics.
- Give suitable references for the calculation of similarity factor in equation (7).
- Please compare the solubility value of dexamethasone with its literature value.
- References: most of the references in reference list are outdated. Authors are suggested to include latest references in the list.
Author Response
Dear Reviewer
We are pleased to consider following points reported during your reviewing process from our manuscript “Mucoadhesive poloxamer-based hydrogels for the release of HP-β-CD-complexed dexamethasone in the treatment of buccal diseases” submitted to the Journal Pharmaceutics Special Issue “Mucoadhesive Drug Delivery Systems: Applications, Challenges and Future Perspectives”.
Point 1: Authors have well prepared and characterized the hydrogels. However, in vivo studies are missing in the article. The inclusion of in vivo data would enhance the validity of the developed formulations.
Indeed, in vivo studies were considered at the beginning of our research. However, economical, social and health-care context and lack of financial support blocked the evaluation of hydrogel therapeutic efficacy in animals. That is why in vivo studies are missing in the article.
Point 2: Abstract: The quantitative information is completely missing in abstract. Kindly included some quantitative information to enhance the readability of the manuscript.
We have included some quantitative information, such as hydration ratio (between 0.218 and 0.191) and poloxamer concentration (from 15 to 17 wt%) used to prepare hydrogels and the prolonged hydrogel retention time (up to 45 minutes) achieved when xanthan gum was added to the hydrogel formulation
Point 3: Introduction: In general, the introduction section is too lengthy. Authors are advised to remove general and well known information from this section.
General and well known and widely described information in scientific literature was removed from this section. You could see the removed information by following the section "Track Changes" in the Word file
Point 4: The symbols in all equations must be in italics, while their subscripts and superscripts should not be in italics.
The symbols in all equations have been changed to Italics. However, for a better readability and legibility, the subscripts were maintained in Italics. You can follow all these changes in the section “Track Changes”
Point 5: Give suitable references for the calculation of similarity factor in equation (7).
We have included two more references for the calculation of similatity factor. These references are:
- Liu J-P, Ma M-C, Chow S-C. Statistical Evaluation of Similarity Factor f2 as a Criterion for Assessment of Similarity Between Dissolution Profiles. Drug Information J 1997;31(4):1255–1271.
- Stevens RE, Gray V, Dorantes A, Gold L, Pham L. Scientific and Regulatory Standards for Assessing Product Performance Using the Similarity Factor, f2. AAPS J 2015;17(2):301–306.
Point 6: Please compare the solubility value of dexamethasone with its literature value.
The solubility value of free dexamethasone acetate (not complexed into cyclodextrin cavity) was compared to those reported in scientific literature. Bibliographic references were added to the manuscript. The values found are quite close to those found in our research study
Point 7: References: most of the references in reference list are outdated. Authors are suggested to include latest references in the list.
New updated references were added to the manuscript. However, and almost concerning mucoadhesion aspects, authors had prefered to refer to first publications well describing mucoadhesion phenomenon and no the lastest reviews resuming them.
We do hope that you reconsider this submission to the Pharmaceutics Journal.
Looking forward to your answer,
Best regards,
Dr. Raul DIAZ-SALMERON
Reviewer 2 Report
The researchers of this study have developed poloxamer P407-based hydrogels containing a poorly water soluble anti-inflammatory drug encapsulated into the cavity of HP-β-CD, with improved mucoadhesive behavior with the aid of excipients such as xanthan gum and sodium alginate. The in vitro drug release studies indicated controlled release of the drug. The work is significant as it opens the way to a new approach of mucoadhesive sprayable hydrogels design for the treatment of mucosal diseases with water-insoluble drugs.
My overall recommedation is accept at present form.
Author Response
Dear Reviewer
We are delighted to learn that you have considered the submission of our manuscript “Mucoadhesive poloxamer-based hydrogels for the release of HP-β-CD-complexed dexamethasone in the treatment of buccal diseases” submitted to the Journal Pharmaceutics Special Issue “Mucoadhesive Drug Delivery Systems: Applications, Challenges and Future Perspectives”.
We are glad to hear that your overall recommendation is to accept the manuscript at present form.
Best regards,
Dr. Raul DIAZ-SALMERON
Reviewer 3 Report
The manuscript "Mucoadhesive poloxamer-based hydrogels for the release of HP-β-CD-complexed dexamethasone in the treatment of buccal diseases" contains interesting findings. However, the following issues must be addressed:
- Introduction: Correct the statement "...use of oral corticoids averting from the severe side effects [25]. Correct the statement: ...disease affecting the most frequently the skin and oral mucosa areas, and particularly involving...
Correct the statement: ...limits the treatment compliance and makes it difficult to reach optimal effects....
2. The introduction should be revised by indicating thermosensitive hydrogels that have been designed for buccal drug delivery.
3. There are grammatical errors which must be addressed by the authors.
4. The novelty of the research should be clearly stated.
5. The preparation of the hydrogels is not clear and must be revised. There is a pressing need to include a comprehensive table for the hydrogels prepared (indicating the composition and amount) and the prepared hydrogels should be given sample names e.g. a,b,c,d, etc.
6. The manuscript is very difficult to read because the hydrogels do not have sample names/labelled making it difficult to understand which of the hydrogels were characterized.
7. The authors should resubmit the manuscript. I will be available to review the resubmitted manuscript. I was unable to read the result and discussion section because the hydrogels were not clearly labelled.
Author Response
Dear Reviewer
We are pleased to consider following points reported during your reviewing process from our manuscript “Mucoadhesive poloxamer-based hydrogels for the release of HP-β-CD-complexed dexamethasone in the treatment of buccal diseases” submitted to the Journal Pharmaceutics Special Issue “Mucoadhesive Drug Delivery Systems: Applications, Challenges and Future Perspectives”.
Point 1: Introduction: Correct the statement "...use of oral corticoids averting from the severe side effects [25]. Correct the statement: ...disease affecting the most frequently the skin and oral mucosa areas, and particularly involving...
Correct the statement: ...limits the treatment compliance and makes it difficult to reach optimal effects....
All the suggested statements were corrected and included in the manuscript text. You could follow all the modifications in the section "Track Changes" of the Word file.
Point 2: The introduction should be revised by indicating thermosensitive hydrogels that have been designed for buccal drug delivery.
Five more references concerning the existing thermosensitive hdrogels for buccal drug delivery have been included in the introduction of the manuscript. These references are:
- Zeng N, Dumortier G, Maury M, Mignet N, Boudy V. Influence of additives on a thermosensitive hydrogel for buccal delivery of salbutamol: Relation between micellization, gelation, mechanic and release properties. International Journal of Pharmaceutics 2014;467(1–2):70–83.
- Zeng N, Seguin J, Destruel P-L, Dumortier G, Maury M, Dhotel H, Bessodes M, Scherman D, Mignet N, Boudy V. Cyanine derivative as a suitable marker for thermosensitive in situ gelling delivery systems: In vitro and in vivo validation of a sustained buccal drug delivery. International Journal of Pharmaceutics 2017;534(1–2):128–135.
- Yu S, Zhang X, Tan G, Tian L, Liu D, Liu Y, Yang X, Pan W. A novel pH-induced thermosensitive hydrogel composed of carboxymethyl chitosan and poloxamer cross-linked by glutaraldehyde for ophthalmic drug delivery. Carbohydrate Polymers 2017;155:208–217.
- Zeng N, Mignet N, Dumortier G, Olivier E, Seguin J, Maury M, Scherman D, Rat P, Boudy V. Poloxamer bioadhesive hydrogel for buccal drug delivery: Cytotoxicity and trans-epithelial permeability evaluations using TR146 human buccal epithelial cell line. International Journal of Pharmaceutics 2015;495(2):1028–1037.
- Al Sabbagh C, Seguin J, Agapova E, Kramerich D, Boudy V, Mignet N. Thermosensitive hydrogels for local delivery of 5-fluorouracil as neoadjuvant or adjuvant therapy in colorectal cancer. European Journal of Pharmaceutics and Biopharmaceutics 2020;157:154–164.
Point 3: There are grammatical errors which must be addressed by the authors.
The final manuscript has been verified twice by an English native speaker. Could you indicate us the gramatical errors that should by addressed by the authors?
Point 4: The novelty of the research should be clearly stated.
As requested, the novelty of the research has been clearly stated in the introduction of the manuscript as you could see on the two following statements:
"The main novelty of this study was the use of cyclic oligosaccharides to improve DMA solubility into a hydrogel..."
"This paper describes for the first time the design of a novel thermosensitive hydrogel for buccal drug delivery intended to be sprayable (presenting a dynamic viscosity of less than 200 mPa.s at room temperature in order to suit for the identified device) and exhibiting an optimized sol-gel transition temperature (30-32 °C). That could allow the in situ gelation once administered in the oral cavity for the treatment of inflammatory diseases ..."
Point 5: The preparation of the hydrogels is not clear and must be revised. There is a pressing need to include a comprehensive table for the hydrogels prepared (indicating the composition and amount) and the prepared hydrogels should be given sample names e.g. a,b,c,d, etc.
A new table (Table 1) has been included in the section "Preparation of hydrogels". This table summarizes the quantitative and qualitative hydrogel composition and given sample names in order to better understand the manuscript.
Point 6: The manuscript is very difficult to read because the hydrogels do not have sample names/labelled making it difficult to understand which of the hydrogels were characterized.
The Table 1, giving sample names to hydrogel preparation and describing qualitative and quantitative composition of hydrogels, could clarify the manuscript making it easier to understand.
Point 7: The authors should resubmit the manuscript. I will be available to review the resubmitted manuscript. I was unable to read the result and discussion section because the hydrogels were not clearly labelled.
As previousl described, the Table 1 could help you to better understand our manuscript
We do hope that you reconsider this submission to the Pharmaceutics Journal.
Looking forward to your answer,
Best regards,
Dr. Raul DIAZ-SALMERON
Round 2
Reviewer 1 Report
Authors have addressed the previous concerns. The new version of this manuscript is suitable for the publication in its present form.
Author Response
Dear Reviewer
We are delighted to learn that you have considered the submission of our manuscript “Mucoadhesive poloxamer-based hydrogels for the release of HP-β-CD-complexed dexamethasone in the treatment of buccal diseases” submitted to the Journal Pharmaceutics Special Issue “Mucoadhesive Drug Delivery Systems: Applications, Challenges and Future Perspectives”.
We are glad to hear that your overall recommendation is to accept the new version of the manuscript for the publication in its present form.
Best regards,
Dr. Raul DIAZ-SALMERON
Reviewer 3 Report
- Figure 8 should be correctly labeled using A-K.
- Figure 5 should be correctly labeled using A-K.
- Figure 7 should be correctly labeled using A-K.
- In the discussion section it is appropriate to use the labels A-K.
Author Response
Dear Reviewer
We are pleased to consider following points reported during your reviewing process from our manuscript “Mucoadhesive poloxamer-based hydrogels for the release of HP-β-CD-complexed dexamethasone in the treatment of buccal diseases” submitted to the Journal Pharmaceutics Special Issue “Mucoadhesive Drug Delivery Systems: Applications, Challenges and Future Perspectives”.
Point 1, 2 and 3:
- Figure 8 should be correctly labeled using A-K.
- Figure 5 should be correctly labeled using A-K.
- Figure 7 should be correctly labeled using A-K
Figures 8, 5 and 7 were correctly labelled using A-K as requested in your Review Report. You could follow all the modifications in the section "Track Changes" of the Word file.
Point 4: In the discussion section it is appropriate to use the labels A-K.
As requested during your Review Report, in the discussion section, hydrogel formulations were appropriated labeled using A-K in order to clarify and better understand the manuscript, as you could see in the two following statements:
"The higher elasticity (G’ values at 37 °C) of P407-based hydrogels containing xanthan gum (samples J and K) could explain this difference, because the possibility of micelles to interact with mucin compounds would be increased. Furthermore, the increased viscosity at 37 °C of xanthan gum formulations, compared to sodium alginate (samples H and I) and Carbopol C971P (sample F), would increase the contact time between mucin and mucoadhesive agent, leading to a stronger mucoadhesive behavior..."
"Our strategy was to incorporate mucoadhesive polymers such as Carbopol C971P (sample F), sodium alginate (samples H and I) or xanthan gum (samples J and K) in order to slow down the release of DMA..."
You could follow all the modifications in the section "Track Changes" of the Word file.
We do hope that you reconsider this submission to the Pharmaceutics Journal.
Looking forward to your answer,
Best regards,
Dr. Raul DIAZ-SALMERON